# Assessing the size and growth of the US wetland and stream compensatory mitigation industry

Todd K. BenDor[1]*, Joungwon Kwon[1], T. William Lester[2]

1 Department of City and Regional Planning and UNC Institute for the Environment, University of North Carolina at Chapel Hill, Chapel Hill, NC, United States of America, 2 Department of Urban and Regional Planning, San Jose State University, San Jose, CA, United States of America

* bendor@unc.edu

**Data Availability Statement:** Data cannot be shared publicly because of restrictions on data access imposed by University of North Carolina at Chapel Hill Institutional Review Board. CB 7097 720 Martin Luther King Jr. Blvd. Bldg # 385,

## Abstract

Interest has focused on quantifying the size and scope of environmental markets, particularly those that offset ecosystem impacts or restore natural infrastructure to improve habitat or promote clean air and water. In this paper, we focus on the US wetland and stream compensatory mitigation market, asking: what types of firms make up the mitigation "industry"? What are the economic impacts–i.e., the "size"–of the mitigation industry? How has this industry changed over time? We present the results of a national survey of mitigation firms and construct an input-output model of the industry's economic impacts and employment. We also develop a comparative, 2014 model of the industry using data from a previous study of the broader, ecological restoration economy. Our findings suggest that the (2019, pre-COVID) mitigation industry collects annual revenues (direct economic impacts) in excess of $3.5 billion, which, along with additional *indirect* (supply chain) and *induced* (spillover) economic impacts, combine to over $9.6 billion in total output and support over 53,000 total jobs. We estimate 2014–2019 growth of ~35.2 percent in revenues, ~32.6 percent in total economic impacts, and a compound annual growth rate (CAGR) of 5.25%. This places the mitigation industry within the range of other, well-established industries within the technical services sector. We suggest establishing North American Industry Classification System (NAICS) codes specifically for ecological restoration and mitigation firms, an essential step in generating accurate and consistent employment estimates in the future, particularly at sub-national geographic scales.

## 1. Introduction

A variety of efforts have sought to quantify the size and scope of market mechanisms aimed at restoring or offsetting damage to ecosystems or providing infrastructure that promotes clean air, water, or habitat (e.g., [1, 2]). Particular interest has centered on the world's oldest operating ecosystem service market, the US stream and wetland compensatory mitigation market, created to address evolving requirements for protection of the physical, biological, and chemical features of waterways under Section 404 of the US Clean Water Act (33 U.S. Code §1251 et

Second Floor Chapel Hill, NC 27599-7097 Ph: 919-966-3113 Fax: 919-966-7879.

**Funding:** Funding for this study was provided to T. K. by the Ecological Restoration Business Association (ERBA). The funder had no role in study design, data analysis, decision to publish, or preparation of the final manuscript. The funder – an industry association of ecological restoration firms – provided contact information for their members and the executive director and executive boards (2021 and 2022) provided survey pre-testing input and background information.

**Competing interests:** The authors have declared that no competing interests exist.

seq.). Mitigation now operates as a model for many emerging environmental markets around the world, with regulators applying aspects of the market's administrative and regulatory structures to diverse policy interventions, including sea turtle capture in Peru [3], shark conservation [4], and Lesser Prairie Chicken habitat conservation in the US Great Plains [5].

In this paper, we ask: what types of firms make up the US wetland and stream mitigation industry? What are the economic impacts–i.e., the "size"–of the mitigation industry? Furthermore, how has this industry changed over time? Over the last 15 years, scholars have explored many facets affecting this market's growth and operation, including the barriers of entry and operational risks experienced by mitigation firms, such as those posed by environmental and regulatory variability in creating, permitting, and operating mitigation sites [6–8]. Extensive, politically-induced regulatory uncertainty also arises from a long history of widely varying interpretations of the legal definition of "Waters of the United States" (WOTUS; [9–11]), which determines the extent of the Clean Water Act's jurisdiction over aquatic systems, and therefore, the demand within wetland and stream compensatory mitigation markets [see 12, 13].

Despite these risks, the mitigation industry has undergone considerable growth and transformation over the last decade; the Environmental Law Institute [14–16] has tracked growth and changes in the use of different mitigation techniques. *Mitigation banking* has become the federally-preferred approach for providing offsets (see [17]), whereby dedicated firms ("bankers" or "bank sponsors") speculatively restore large swaths of wetlands, or reaches of streams ("banks"), to generate "mitigation credits" (units of functional uplift), which are sold to land development interests impacting regulated wetlands and streams (i.e., transferring liability for providing compensatory mitigation from the impactor to the banker). In 1992, the Environmental Law Institute [14] found only 46 approved banks, nationwide. This figure grew nearly fivefold to 219 banks in 2002 [15], 405 in 2005 [16], 1,800 in 2013 [18], and nearly 2,200 by October 2022 [19].

There have been many efforts to measure the economic activities of the mitigation market. Most widely referenced has been the Environmental Law Institute's (2007; [20]) estimate of total annualized spending on Clean Water Act Section 404 compensatory wetland mitigation during US Fiscal Year 2003 (Oct 2002-Sept 2003), which totaled approximately $2.9 billion (the higher value of $3.8 billion is often erroneously cited as the "economic size" of the mitigation industry, but this figure included mitigation performed under additional regulatory frameworks and ecosystem protections (e.g., Section 10 of the US Endangered Species Act). The authors derived this estimate by multiplying the average price of mitigation (per acre, disaggregated by geographic area and mitigation method; gathered via a survey of regulators) by the total mitigation performed (based on regulator data).

Using a similar approach (and armed with gradually improving data, eventually disaggregated into wetland and stream transactions), researchers at Ecosystem Marketplace, Inc.–a web-based information source for environmental finance and markets–succeeded the Environmental Law Institute's [20] efforts, repeatedly estimating the transactional volume and value of the mitigation market as part of broader efforts to assess the state of watershed and green infrastructure investments [21–23], private investment in conservation [24], and biodiversity markets and mitigation [25–27]. Collectively, these efforts estimated the transactional volume of wetland and stream credits at $1.3 - $2.2 billion in 2010 [25], $1.8 - $3.2 billion in 2011 [26] and $1.45 - $5.7 billion in 2017 [27].

In 2014, BenDor et al. [28] estimated the economic *impacts*–i.e., impacts on employment and economic output, as opposed to economic *benefits*, i.e., the economic impacts of improved ecosystem services or outcomes–of the broader, US "ecological restoration economy," of which the mitigation industry is a part. The authors found that restoration activities employed ~126,000 workers and generated $9.5 billion in revenue, annually. While this was the first

national estimate of its kind, a concurrent review by BenDor et al. [29] highlighted a large number of economic impact studies performed for locally-focused (or agency-specific) ecological restoration programs. However, they found few (if any) economic impact studies accounted specifically for mitigation activities (e.g., see [30], which use similar methods to our study).

Assessments of industry-specific economic activity and employment typically rely on standard industrial classifications schemes (e.g., the US Census Bureau's North American Industry Classification [NAICS] codes; [31]) and public data sources. Unfortunately, firms that perform wetland and stream mitigation–accomplished primarily through environmental construction and re-vegetation activities–do not currently have their own industry category and are each classified under a variety of existing industry categories, such as environmental consultancies, earth movers, real estate or environmental law firms. These mis-classifications make it difficult to estimate the accumulated employment or economic activities and impacts of these firms. However, these firms now collectively represent a cohesive and well-defined industry, which is, itself, part of the rapidly growing US ecological restoration sub-sector and green technology sector.

In this paper, we present the findings of a 2021 national survey of firms that perform wetland and stream mitigation, aimed at determining the "current" (2019) makeup of the industry and its employment and economic impact characteristics. Using data from BenDor et al.'s [28] 2014 analysis of the economic impacts of the ecological restoration sub-sector more broadly, we also retroactively estimate the 2014 subset of the restoration economy involved in wetland and stream mitigation activities and compare it to 2019 values.

Our findings suggest that the (2019, pre-COVID) mitigation industry receives annual revenues (direct economic impacts) in excess of $3.5 billion, while creating additional *indirect* (i.e., supply chain) and *induced* (i.e., spillover) economic impacts of ~$2.3 billion and ~$3.8 billion, respectively. In total, the industry produces a total economic impact of over $9.6 billion (these values do not include several billions in economic value from the environmental *benefits*–i.e., improved ecosystem services and outcomes–of mitigation sites). While we also find that this economic output supports over 53,000 jobs, nationwide, our analysis suggests that establishing specific NAICS codes for the mitigation industry will become essential for accurate employment estimates, particularly at more localized (sub-national) geographic scales.

## 2. Methods and data

Relying on past efforts to evaluate the ecological restoration [28] and green infrastructure industries [23] as a framework, assessing the economic impacts of the US wetland and stream mitigation industry requires four key steps. First, we must characterize the universe of firms involved in the industry; what is the complete set of firms that perform mitigation-related activities? Second, we need to survey these firms to collect information about their revenue, specifically the subset of revenue from mitigation-related activities (many firms also perform non-mitigation work).

Third, as surveys–especially those asking sensitive questions about a firms' income–rarely garner full, 100% response rates, we must make assumptions about how well responses represent the universe of mitigation firms as a whole (i.e., do those firms that *did* respond represent those that *did not* respond?). Fourth, and finally, we need to understand how funds paid to mitigation firms (i.e., payments for mitigation credits; "direct impacts") flow through the economy; mitigation firms' revenue flows to other firms through supply chains (*indirect* impacts; e.g., payments to nurseries, earth movers, engineers) and as a result of wages paid to employees (e.g., buying groceries, staying in hotels; *induced* impacts).

## 2.1 Establishing a universe of mitigation firms

A convenient (and ultimately, conservative) way that we can characterize the full extent of an industry draws on established industry associations and their memberships. In 2014, the mitigation industry was represented in national policy discussions by the National Mitigation Banking Association (NMBA). Established in 1998, the NMBA "promote[d] federal legislation and regulatory policy that encourage[d] mitigation banking [under the US Clean Water Act] and conservation banking [under the US Endangered Species Act] as a means of compensating for adverse impacts to our nation's environment. This commitment [was] fulfilled through a variety of research, education, and outreach programs sponsored by the NMBA and available exclusively to its members [32]." In late 2017, some members of the NMBA established a new organization, the National Environmental Banking Association (NEBA), to solely focus on representing the interests of mitigation banking firms [33]. Earlier that same year, the remainder of NMBA re-branded as the Ecological Restoration Business Association (ERBA) [34], which is focused on representing firms involved in many forms of ecological restoration, including, but not limited to, wetland and stream mitigation banking.

ERBA provided three, comprehensive membership lists used for this study, including: 1) a list of all current member firms, including primary (and occasionally, secondary) contacts within those firms (81 unique firms; n = 203 individual contacts; Table 1), 2) a list of lapsed/former member firms (including firms that had left the industry, gone out of business, or merged with other firms; 87 unique firms; n = 200 individual contacts), and 3) a separate list of non-member firms in California that are members of CalERBA [35], a nascent California state association of restoration firms (that grew out of ERBA, but is a separate legal entity; 4 additional unique firms; n = 7 individual contacts).

Due to their internal privacy policies, NEBA was unable to share their membership list. However, we were able to obtain a contact list of five years (2017–2021) of attendees at the industry's primary professional conference (the National Mitigation and Environmental Markets [formerly Ecosystem Banking] Conference; [36]), where NEBA members, as well as upstart and mitigation industry-adjacent firms, are in near-universal attendance. After removing duplicates of ERBA's contacts, this list represented another 328 firms (n = 470 individual contacts). Additionally, six firms were suggested via a snowball question (asking respondents to suggest additional firms to contact regarding this survey), but all were already on one of the four existing contact lists.

## 2.2 Surveying mitigation firms

In order to determine mitigation firms' revenue and employment (which collectively represents the "direct impacts" of the industry), we adapted the web survey implemented by BenDor et al. [28] to assess the size of the broader ecological restoration industry. The University of

**Table 1. Mitigation industry contact universe and survey response rates for each list.** Seven duplicate responses from six firms were removed, as well as one response that did not list a firm.

| | Contacted | | Responses | | | |
|---|---|---|---|---|---|---|
| | Individuals | Firms | Individuals | Firms | Firms performing mitigation (restoration) | Adjusted firm response rate |
| ERBA | 203 | 81 | 34 | 28 | 22 (28) | 34.6% |
| ERBA Lapsed | 200 | 87 | 5 | 5 | 4 (4) | 5.7% |
| NMEMC | 469 | 328 | 50 | 49 | 33 (48) | 14.9% |
| CalERBA | 7 | 4 | 2 | 1 | 1 (1) | 25.0% |
| Total | 879 | 500 | 91 | 83 | 60 (81) | 16.6% |

North Carolina at Chapel Hill's Institutional Review Board approved this survey (designation as Non-Human Subjections Research [NHSR]; UNC IRB #13–1872), and we obtained written consent from all respondents, guaranteeing response confidentiality. We refer readers to S1 File for details on survey modifications, pre-testing, and implementation for the current study.

It is important to note that we asked firms to base their responses on restoration sales and activities from 2019 to avoid incorporating the impacts of the COVID-19 pandemic, which evidence suggests had widely varying (and regionally specific) impacts on infrastructure construction (e.g., [37]), a major source of mitigation demand and therefore mitigation sales (S. Johnson, ERBA Executive Director, personal communication). While this stipulation required firms to track down "historical" data on their business activities (this survey was implemented in late-2021), it also allowed us to evaluate the five year change (2014–2019) in the mitigation industry. See S2 File for information on our approach to survey weighting (inverse response probability).

## 2.3 Input-output modeling

Input-output (I-O) modeling is a method for calculating how changes in the demand for work in a given sector–i.e., wetland and stream mitigation–impact the broader economy. This technique has been found to be a useful way to measure net changes in economic activity as a result of funding for development or infrastructure expansion [38]. I-O modeling has been used in a wide variety of applications, including efforts to measure the economic impacts of federal, state, and local regulations on industries and the localized economic impacts of transportation projects, government investments, and private investments [39]. I-O modeling has also been previously used to measure the economic impacts of investments in the broader "ecological restoration economy" [28, 40].

In an I-O model, economic impacts accrue from the *direct effects* of an investment in a firm–in our case, the payments made to mitigation firms as a requirement for obtaining impact permits represent additional "final demand" (i.e., the value of goods and services sold to end users) for the mitigation industry. The total, industry-wide values of these payments are derived from our survey of mitigation firms. These payments represent mitigation firm revenues and pass through the firm to create *indirect* and *induced effects* of mitigation.

In order to accommodate increased mitigation demand, these payments create *indirect effects* (i.e., changes in demand in the "backward linked" industries that form the supply chain for mitigation firms and create the goods and services purchased by mitigation firms to create restoration sites). In other words, in order to complete a mitigation project, firms must purchase materials and services from other firms, such as plants to vegetate project areas or environmental engineering services. Because of these purchases, firms in related sectors also experience economic activity as a result of investment in mitigation. Thus, those related sectors are activated indirectly as a result of the direct sales of mitigation firms.

Finally, the *induced impacts* of a mitigation investment are the changes in household spending as a result of wage changes in *directly* and *indirectly* affected industries. Payments to mitigation firms result in payments to their workers–as well as the workers throughout the mitigation supply chain–who spend money throughout the economy. The *total* economic impact is found by summing direct, indirect, and induced impacts.

While we derive the *direct* impacts of the mitigation industry from our survey, we used IMPLAN v.3.1 (IMpacts for PLANning; Minnesota Implan Group; [41])–an industry-standard I-O modeling package and data platform–to calculate the *indirect* and *induced* impacts. IMPLAN uses data–updated every five years–from the US Bureau of Economic Analysis and other federal statistical agencies to describe the purchasing relationships between all sectors of

the US economy (or smaller geographies). The US Census [31] classifies these industries hierarchically, designating individual NAICS industries (giving them 5-digit NAICS codes) within industry groups (4-digit), which are within subsectors (3-digit), which are within sectors (2-digit). Industry groups–which have 4-digit NAICS codes–are the most detailed industry classification for which there is consistent, publicly-available data. We use IMPLAN's 2019 structural matrices for the entire United States for all analyses and all results are reported as USD2021.

Any I-O analysis inherently makes a number of assumptions. For example, we do not consider "forward linkages" between producers and consumers (e.g., new firms as a result of changes in mitigation sales). We also assume that suppliers for direct and indirect inputs are providing their goods and services from standing inventories, which would not be used elsewhere in the economy (at least to the extent that it would result in a price change). By focusing on readily quantifiable economic outcomes, such as jobs, value-added, and output, we are evaluating a narrow scope of economic impacts that result from restoration. It is important to note that this type of study does not account for the economic *benefits* of restored ecosystems that accrue from improved ecosystems goods and services (e.g. flood prevention; [42]).

## 2.4 Assessing industry growth: Post-hoc comparison to 2014 data

In order to estimate how the mitigation industry has changed between 2014 and 2019, we require a method of defining and "backcasting" the activities of the industry in 2014; what counts as "mitigation," as opposed to other ecological restoration activities? What firms should characterize the universe of firms involved in the industry in 2014? While the 2014 survey *did* collect information about firms' mitigation industry-involvement, unfortunately, these questions were not structured in ways that allow us to retroactively estimate the proportion of firms' revenue from their mitigation activities.

S3 File contains details on BenDor et al.'s [28] sampling approach and characterization of the mitigation industry, as well as our evaluations of different response sub-sets that allow 2014 to 2019 industry comparisons. These evaluations reveal a subset of firms (n = 112 firms) that comprise a reasonably complete set of responses, representative of the mitigation industry at that time. Moreover, this 2014 firm subset can be replicated in our 2021 survey (see S4 File for the full 2021 survey instrument) by analyzing only firms involved in either Section 404 mitigation (Q15; n = 51 firms) or in mitigation banking (Q23; n = 56 firms). This selection reduces the 2019 sample of "mitigation-involved" firms to n = 60 firms. For these firm subsets, we calculate the margins of error associated with each survey at 9.3 percent (2014) and 12.7 percent (2019), respectively (see S3 File for more details).

Throughout the remainder of this paper, we will describe the 2019 mitigation industry (data collected via 2021 survey) in terms of this subset of firms, as well as use this subset to compare the 2019 industry to its 2014 counterpart. Additionally, we will use the same 2019 IMPLAN structural matrices in order to represent the same structure of the economy for both time periods.

## 3. Results

### 3.1 Describing the 2019 mitigation industry

We asked respondents to list their primary industry, selecting the NAICS code that best described their firm's activity. Using these selections, we calculated the weighted sales related to restoration for each industry listed. Table 2 lists the sales figures by NAICS for the top 10 industries/sectors reported, which represent the vast majority of sales (99.0 percent).

**Table 2. Top 10 current NAICS-defined industries within the 2019 mitigation industry by estimated sales.**

| NAICS Code | Weighted Sales, 2019 (USD2021) | % of total | Cumulative % of total |
|---|---|---|---|
| 5413-Architectural, Engineering, and Related Services | $ 1,292,497,129.67 | 39.4% | 39.4% |
| 5416-Management, Scientific, and Technical Consulting Services | $ 500,212,108.92 | 15.3% | 54.7% |
| 8133-Social Advocacy Organizations | $ 345,752,727.27 | 10.6% | 65.3% |
| 2389-Other Specialty Trade Contractors | $ 334,752,992.00 | 10.2% | 75.5% |
| 2379-Other Heavy and Civil Engineering Construction | $ 310,976,417.82 | 9.5% | 85.0% |
| 5419-Other Professional, Scientific, and Technical Services | $ 173,777,943.34 | 5.3% | 90.3% |
| 1153-Support Activities for Forestry | $ 136,181,447.27 | 4.2% | 94.4% |
| 5417-Scientific Research and Development Services | $ 62,727,272.73 | 1.9% | 96.4% |
| 5239-Other Financial Investment Activities | $ 51,854,545.45 | 1.6% | 97.9% |
| 5411-Legal Services | $ 34,374,545.45 | 1.0% | 99.0% |

We received responses from firms in 23 states, with respondents most frequently headquartered in North Carolina (10), California (9), Texas (6), and Colorado (4). While firms were most commonly headquartered in these states, they may be working in numerous others. The median firm was started in 2005, a decrease in age from 2014's median founding year, 1994. The median firm also has 13 full-time employees (and 4 part-time employees); approximately 71% of firms have under 100 full-time employees, thereby qualifying as small businesses (see 13 CFR 121, pgs. 3280–3304). The educational attainment of mitigation firms' employees is high, with over 79.5 percent having received bachelors or graduate degrees (Fig 1A).

The median respondent firm received $1–5 million in annual revenue (Fig 1B). In terms of revenue change, 73.3 percent of respondents stated that, over the last five years, their firms' total revenue (sales) had increased, while 10 percent stated that their revenue had decreased. Breaking down the sources of this revenue, 45.0 percent of respondents saw an increase during this period in the portion of this revenue that came from ecological restoration work (although this may not just be mitigation), while another 45.0 percent saw this portion remain the same.

These firms engaged in a wide range of activities, with the most common being monitoring, operations, and maintenance, as well as consulting service, and planning and permitting services (a new response option, not previously available in BenDor et al.'s [28] survey; Fig 1C). Mitigation firms work in a wide variety of ecosystems; while most work in freshwater ecosystems (performing wetlands and stream/riparian restoration), many others restore forests, saltwater wetlands, grasslands and coastline/marine ecosystems (Fig 1D).

In terms of legal drivers (Fig 1E), along with Section 404 of the Clean Water Act, we find that many firms have engaged in habitat mitigation/restoration required under the US Endangered Species Act [43], as well as water quality restoration and/or nutrient trading activities prompted by Section 402 of the Clean Water Act [44]. Along with the federal drivers, 45 percent of respondents (n = 27) identified state and local legal drivers of their restoration work, including programs in at least 15 states (and the Chesapeake Bay program; [45]). These programs include a variety of legal frameworks and programs, including local stormwater requirements (e.g., Kane, Lake, and DuPage County stormwater ordinances in Illinois; [46]) and municipal separate stormwater sewer system (MS4) programs [47], state-level endangered species rules (e.g., CEQA and Sage Grouse protections in Wyoming), and state-wide nutrient and water quality programs (e.g., Maryland Clean Water Commerce Act; [48] and North Carolina's Nutrient Offset & Buffer Mitigation programs [49].

Finally, we asked respondents about the barriers that affect their firms' ability to obtain more restoration-related work, essentially a question about the emergent factors affecting industry growth. Among responding firms, 35 percent (n = 21) stated that barriers existed and

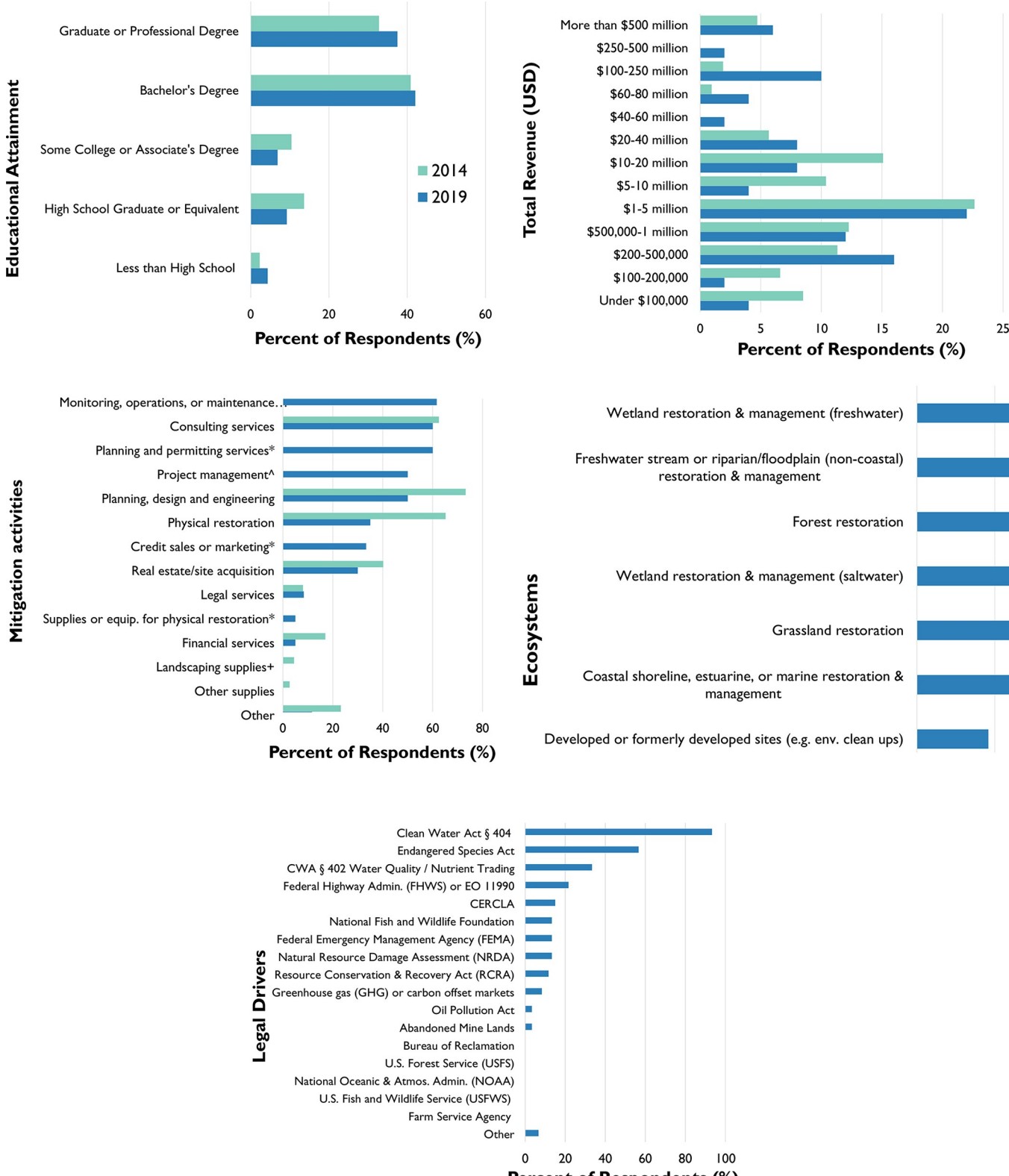

**Fig 1.** Survey summary statistics showing percentage of respondents by category: (A) Employee education levels (2014 and 2019); (B) Firm revenue (2014 and 2019); (C) Mitigation-related activities (2014 and 2019, + indicates choice only available on 2014 survey; * indicates choice only available on 2019 survey); (D) Ecosystems where restoration work was completed (2019); and (E) Federal legal drivers of restoration work (2019).

13 explicitly identified regulatory uncertainty and/or "slow" regulatory processes as the primary barrier. Respondents expanded on this barrier, notably commenting:

"Clarity, consistency, and timeliness in the project approval process is an issue for us, even though the 2008 [Rule; [17]] dramatically improved the situation."

"Regulatory uncertainty due to changes associated with local and national policies and procedures. Limited resourcing, funding, and poor performance implementing mitigation regulations, obstructive comments from [regulator] agencies, lack of leadership."

"There is a huge amount of regulatory uncertainty ([i.e., from] WOTUS), the processing time for permits and approvals is absurd, there is inconsistent enforcement of the laws (Section 404)..."

"It is very unfair that our regulators can change our rules and answer to no one; they are effectively also our competition, as they make it very difficult for [some potential customers] to get approved [to use]...our existing mitigation bank, as rules state. We invested $1 million to develop [our bank]; we have recouped only $250,000 in 13 years of credit sales.... We have developed a letter specifically coaching [potential customers] how to argue with [regulator] to honor [our mitigation bank instrument]!"

Respondents also described instability in demand as an important barrier to growth:

"...[I]nvestment returns are highly variable (lots of sales one year / few sales the next). [This is a] very difficult business."

"Instability of restoration demand due to COVID impacts on state and local government operations."

For some respondents, a combination of regulatory and demand uncertainty has created barriers to receiving investments, particularly for smaller firms:

"[Our] small business and risky industry creates limited conventional funding options."

"[It is] hard to raise capital at our size."

Finally, one respondent described tax uncertainty as a barrier to recruiting landowners whose land could be restored for mitigation purposes:

"Income land owners receive from my company related to purchase of conservation easements is taxed as capital gains and there is worry that federal capital gains tax will increase, providing less incentive for the land owners to place their land into conservation easements for mitigation projects."

### 3.2 I-O modeling of the 2019 mitigation industry

Using the 2021 survey responses as weighted inputs into IMPLAN v. 3.1, we calculated the direct, indirect, and induced effects of the 2019 wetland and stream mitigation industry (Table 3). We estimate that, during 2019, the mitigation industry produced ~$3.54 billion in economic output, including the value of all sales or revenue to firms engaged in all aspects–from planning to planting–of wetland and stream mitigation work (i.e., direct impact). This activity directly generates over 21,000 jobs each year and approximately $1.75 billion in labor

**Table 3. Input-output (I-O) modeling of the economic impacts of the 2019 and 2014 US wetland and stream mitigation industry (all values in USD2021).** Rows sum vertically, but columns do not sum horizontally. "Value added" calculations include labor income and firm profit, while "economic output" additionally includes the costs of inputs (i.e., cost of goods sold).

| | Impact Type | Employment | Labor Income | Value Added | Gross Output |
|---|---|---|---|---|---|
| **2019** | Direct Effect | 21,470.90 | $1,753,601,029 | $2,100,480,648 | $3,541,481,993 |
| | Indirect Effect | 11,156.50 | $781,115,513 | $1,206,924,766 | $2,288,312,824 |
| | Induced Effect | 20,594.10 | $1,208,462,579 | $2,144,066,831 | $3,830,754,906 |
| | **Total 2019 Effect** | **53,221.40** | **$3,743,179,121** | **$5,451,472,245** | **$9,660,549,723** |
| **2014** | Direct Effect | 18,542.50 | $1,357,786,295 | $1,516,386,236 | $2,619,068,604 |
| | Indirect Effect | 8,339.70 | $587,597,727 | $904,468,279 | $1,724,609,070 |
| | Induced Effect | 15,806.50 | $927,526,088 | $1,645,588,142 | $2,940,119,399 |
| | **Total 2014 Effect** | **42,688.70** | **$2,872,910,110** | **$4,066,442,658** | **$7,283,797,073** |

income (i.e., wages and benefits). We can calculate the average labor income per direct job at $81,673.38, which represents a figure well over the 2019 average annual wage in the US ($55,025.88, also in USD2021; see [50]).

Direct spending on mitigation activities results in just over $2.1 billion in value added in the U.S. economy. As noted by BenDor et al. [28], it is important to focus on total output per full-time job as well as the total count of employment, to put this figure in context because this gives a sense of the opportunity cost of the labor employed in mitigation. Our analysis indicates that mitigation activity generates approximately $164,943 in output per job. In total, this would generate nearly $377 million in state and local tax revenue, and $761 million in federal tax revenue, annually.

Beyond the direct impact, the mitigation industry also supports additional employment and economic output through indirect spending (i.e., spending by mitigation firms on inputs) and induced (i.e., household) spending. As indicated in Table 3, the indirect effect represents an additional 11,156 jobs and $2.29 billion in output, while 20,594 jobs and $3.83 billion in output are generated through household spending. All included, we estimate that the wetland and stream mitigation industry generates over 53,200 jobs and over $9.6 billion in economic output. From these numbers we can calculate standard employment multipliers, both per 100 direct jobs created (i.e., the number of non-direct jobs created per 100 direct jobs: 147.9) and per $1 million in final demand (6.1 direct jobs, and 9.0 additional non-direct jobs).

### 3.3 Industry change, 2014–2019

The result of IMPLAN modeling of the activities of mitigation firms in 2014 can be seen in the second set of rows Table 3 (still in USD2021). We estimate that, in 2014, the mitigation industry produced ~$2.6 billion in direct economic output, employing just over 18,500 people and generating $1.36 billion in labor income (i.e., wages and benefits). Nationally, this generated nearly $279 million in state and local tax revenue, and $579 million in federal tax revenue.

This "backcasting" of the industry's activities suggests a 35.2 percent increase in output and a 15.8 percent employment increase over the five year period between 2014 and 2019, as well as increases of ~35.1 percent and ~31.5 percent in total state/local and federal taxes paid by the industry, respectively.

Measuring a 29.2 percent increase in labor income, we can calculate the industry's compound annual growth rate [51]: CAGR = $(V_{final}/V_{initial}^{1/period} - 1)$, to be 5.25 percent. Beyond the industry's direct economic impacts, we also see growth in indirect and induced economic output (32.7 percent and 30.3 percent, respectively) and employment (33.8 percent and 30.3 percent, respectively). In terms of total economic impact (sum of direct, indirect, and induced

effects), we estimate a 2014–2019 employment growth of 24.7 percent and economic output growth of 32.6 percent.

## 4. Discussion

### 4.1 Coverage of estimates and caveats

While lower than some previous restoration survey efforts [52], the response rates garnered by our 2019 survey are well in line with rates seen in other surveys that ask sensitive business questions [53], and higher than rates achieved by the 2014 survey described by BenDor et al. [28]. While we have no well-established (and universally accepted) population of firms to which we can compare our sample, we are confident that our responses are broadly representative, as they include firms that perform different roles throughout the mitigation industry [7].

However, we do note that, by concentrating our survey on members of a restoration industry association and attendees of the primary industry conference, our results are likely conservative. Unfortunately, our method may miss a substantial number of firms that do not identify as being part of the "mitigation industry"–i.e., firms that perform their own mitigation on a case-by-case basis (without contracting with specialized mitigation firms), or organizations that are required to perform so much mitigation (e.g., state transportation departments or the Disney Corporation; [54]), that they develop in-house expertise to leverage ecological restoration's substantial scale economies [46, 55].

### 4.2 Comparison to previous estimates and to other industries

How do our estimates compare with previous efforts to gauge the size and growth of the industry? As prior studies have framed their calculations as efforts to derive the "total costs" or "total spending" associated with stream and wetland mitigation (we have removed all endangered species components), we can directly compare these values with our estimates of the total direct output associated with the industry in 2014 and 2019. Fig 2A shows the estimated ranges of total industry sales in previous studies (inflation-adjusted to USD2021). The comparatively small ranges of our 2014 and 2019 estimates reflect the relatively low rate of total survey errors, as well as the alternate technique that we employ. By virtue of these smaller ranges, our estimates also are novel in clearly demonstrating growth in the industry over a five year period.

The Environmental Law Institute's [20] early work to estimate the size of the industry represents a herculean effort during a period of extremely limited (and often incorrect; [56]) data. Their approach likely included a more comprehensive range of mitigation activities available, including nascent in-lieu fee mitigation programs (administered by local governments and non-governmental organizations; [57]) and "permittee-responsible mitigation," performed by impactors and consultant firms which, at the time, would not likely identify as part of the "mitigation industry" [46]. However, given that their mid-point estimate in 2003 is higher than that of Bennett et al.'s [27] almost 13 years later (again, both inflation-adjusted to USD 2021), we suspect that the Environmental Law Institute [20] may have over-estimated the geographically-specific prices associated with mitigation sales. This is an aspect of the industry that remains notoriously secretive nearly 20 years later [6], as demonstrated by the <20 percent response rates of this our study and that of BenDor et al. [28].

While more recent industry size assessments shown in Fig 2A–which were derived using similar methodologies to those of the Environmental Law Institute [20]–suggest industry growth between 2009 and 2016 (dates of data collection), the relatively large ranges of each of the estimates overlap, considerably. Bennett et al. [27] estimate the industry at $4.09

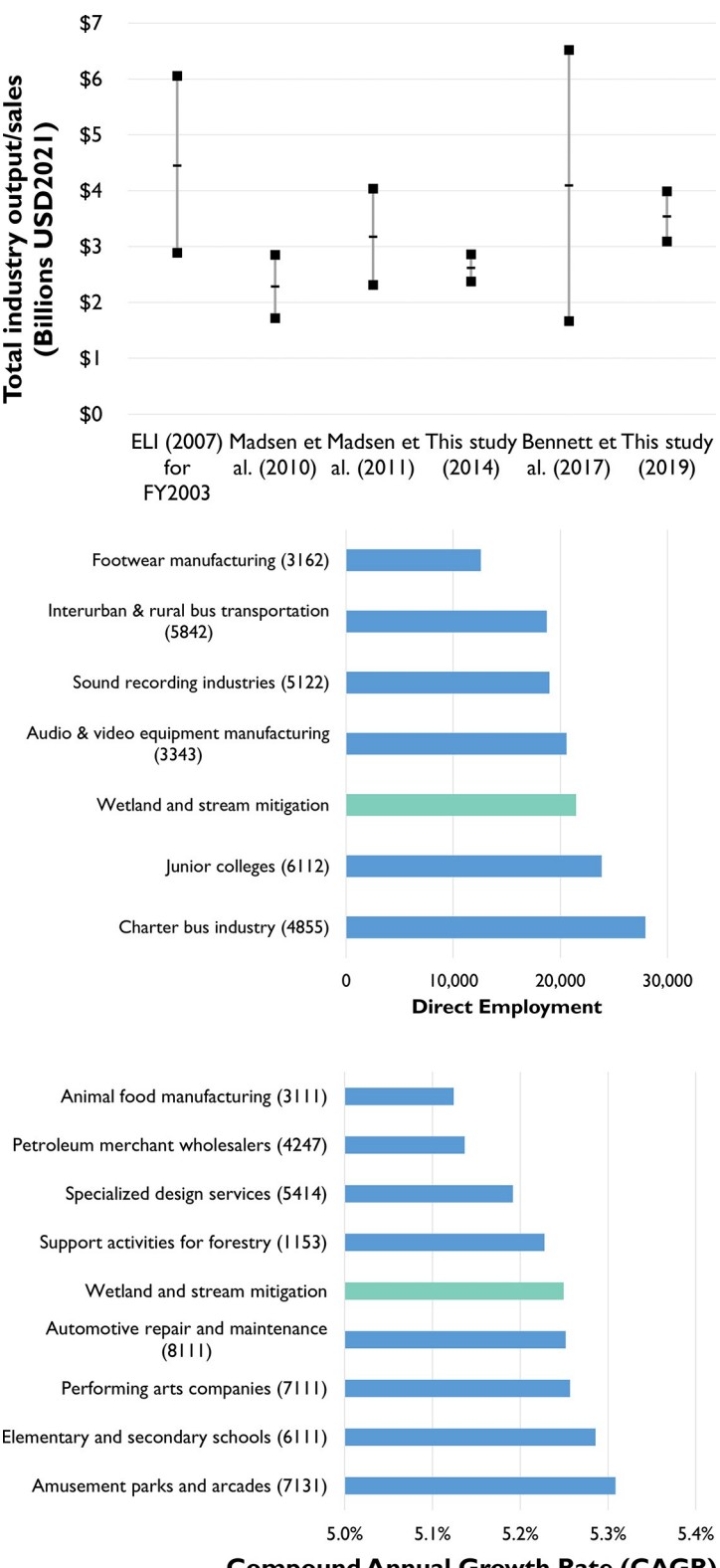

**Fig 2.** (A) Comparison current study with previous wetland and stream mitigation industry size estimates (adjusted for inflation to USD2021). Lines indicate low-end, mid-point, and high-end estimates for each study. (B) Direct employment comparison to with other industries (2019). (C) Compound annual growth rate comparison with other industries (2014–2019). NAICS 4-digit classifications given in parentheses. Data source: (Bureau of Labor Statistics 2022).

billion ± 59.3% (USD2021), a range significantly larger than previous estimates, their study was aided by strongly improved data compared to Madsen et al.'s [25, 26] previous studies.

We can also compare the mitigation industry to other industries in terms of employment and a 5-year growth rate (in terms of labor income). Fig 2B puts the direct employment that we attribute to the mitigation industry in context of several, similarly-sized industries (as classified at the 4-digit NAICS code level). While mitigation employed slightly fewer workers in 2019 than the charter bus and junior college industries, our model suggests that it had grown beyond the sound recording and footwear manufacturing industries. Likewise, among the 301 4-digit NAICS industries for which we can calculate a 2014–2019 compound annual growth rate, we find that the mitigation industry's 5.25% CAGR puts it in the top 36% of all industries (ranking 108 of 302). Fig 2C puts this rate in context with other industries with similar growth rates.

We calculated the average labor income per direct job in the mitigation industry at $81,673.38. It is worth noting that this figure dwarfs the associated value for construction industries and falls in line with that of legal services and environmental and other technical consulting services (Fig 3). Likewise, our analysis indicates that mitigation activity generates approximately $164,943 in output per job. While this figure is lower than some highly capital intensive industries like oil extraction and manufacturing, it is quite a bit larger than the

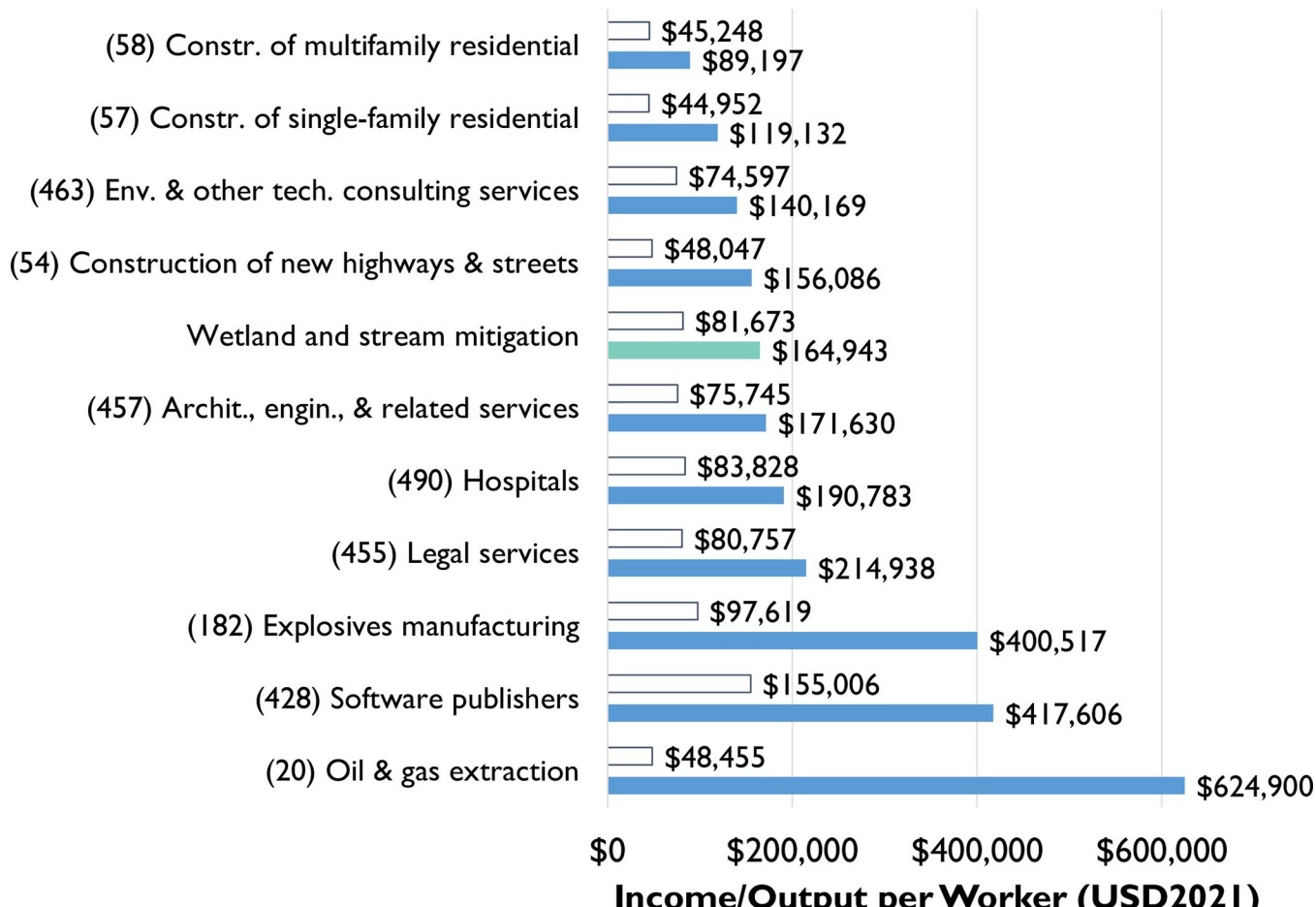

**Fig 3. Comparison of wetland and stream mitigation income per worker (white bars) and output per worker (blue bars) against other industries.** Source: (Bureau of Labor Statistics 2022).

residential construction industry ($89,000 and $119,000 for multi-family and single-family residential, respectively) and slightly larger than highway construction ($156,086), industries that are strong drivers of the wetland and stream mitigation industry.

Finally, we can evaluate the mitigation industry's propensity to create jobs by looking at its "employment multipliers": two common ways to evaluate an industry's propensity to create jobs as a function of demand are to look at 1) the number of non-direct jobs supported by a direct job in an industry (or more commonly, by 100 direct jobs), and 2) the rate at which direct, indirect, and induced jobs are created as a function of a $1 million investment in final demand of an industry (i.e., direct investment in mitigation–taking a $1 million piece of the total direct output calculated from Table 2). While we find that mitigation's first measure–the number of non-direct jobs supported by an industry to direct jobs–is on the lower end of comparable professional, scientific, and technical service industries (i.e., NAICS sub-sector 541; Fig 4A), it is important to note that the industry supports a comparably high number of direct jobs, given a fixed investment (Fig 4B). Moreover, our results suggest that mitigation is comparable to other service industries (and construction) in terms of its total rate of job production.

## 4.3 Evolution of the mitigation industry

It is interesting to see that the average age of companies has decreased slightly since the 2014 survey (median age shifting from 20 years to 17 years). Several dynamics may be at play here; first, as one respondent highlighted, mitigation is "a very difficult business," and firms may regularly go out of business, with newer firms taking their place. Alternatively, mitigation firms have notoriously complex legal structures, with some firms creating separate (limited liability) legal entities for each mitigation project [13]. Depending on whether survey respondents referred to a parent firm or a project-level entity, we may see quite a bit of noise in terms of measuring "firm age."

Conversely, anecdotal evidence has suggested some concentration within the industry (e.g., [58]), as larger environmental consulting and mitigation banking firms have purchased smaller firms to establish footholds in new geographic markets or as an "acqui-hiring" strategy (i.e., acquisitions as a means of hiring talented personnel; [59]). However, if this behavior dominated within the industry, we might expect the average firm size and age to increase over time–which we do not find. Instead, our results suggest that the industry has attracted a significant number of newer firms to enter the market, a factor that may be driven by rapid growth in Environmental, Social and Governance-focused (ESG) investment activity and capital [60].

Finally, it is important to contextualize the observed growth of the industry within the ongoing debate over the extent of Waters of the United States (WOTUS), which–in the absence of consistent, state-level interventions–defines the potential demand for wetland and stream mitigation markets as a result of federal regulation. During our study period, legal interpretations have swung widely as to the geographies and typologies of ecosystems regulated across the United States [61]. Previous efforts [10] have explored the profound impacts that changes in WOTUS interpretations can have on the stability of mitigation demand and, ultimately, on the long-term viability of mitigation firms.

## 5. Conclusions

The key goals of this study's empirical analysis were to a) provide a broadly representative national picture of the economic impacts of the wetland and stream mitigation industry, b) estimate the total sales and employment of mitigation firms, c) describe mitigation activities, and d) describe mitigation firms and how they are currently segmented by standard industrial

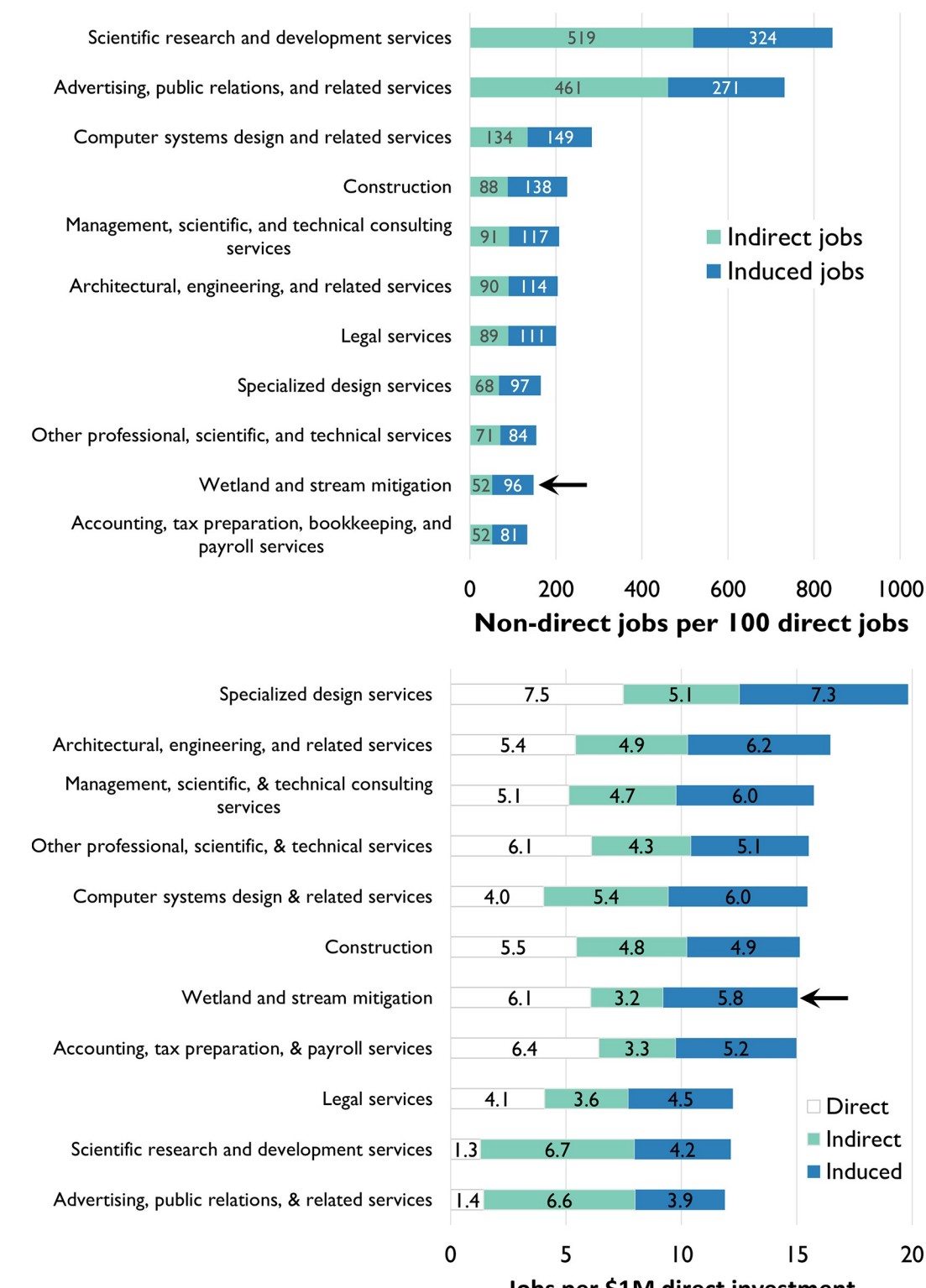

**Fig 4.** Employment multiplier of wetland and stream mitigation and comparable service industries (and construction), in terms of (A) non-direct (indirect and induced) jobs per 100 direct jobs and (B) direct, indirect, and induce jobs supported per $1 million in direct investment. Data source: (Bivens, Josh. *Updated Employment Multipliers for the U.S. Economy*. Economic Policy Institute, 2019. https://www.epi.org/publication/updated-employment-multipliers-for-the-u-s-economy/).

classification measures. In performing this analysis, we set out to better understand the industry and how it has changed in recent years.

We can frame the mitigation industry in its present state–and potential future forms–in terms of a broader discussion of economic development. An important underpinning of our understanding of economic development is rooted in economic base (or export base) theory [62], which is premised on the notion that the "external demand for a region's products is the primary determinant of regional prosperity" (i.e., why the economies of some geographic regions are stronger, more stable, or more poised for growth, than others). Malizia et al. [63] note that, when applying this theory, the economic activities of a metropolitan labor market can be separated into those that produce for export to markets outside the region, and therefore bring income into the region (e.g., "base" industries; while historically viewed as primarily extractive, such as mining or fisheries [64], the evolution of the technology industry in California's Bay Area region, for example, deconstructs the notion that base industries must necessarily be extractive.) and those that produce for the local market ("non-basic" or "service" industries). Under this theory, two linkages prompt the multiplier effects that we discuss above: 1) the base industry directly purchases goods and services from non-base industries (i.e., indirect impacts), and 2) workers in the base industry purchase food, entertainment, clothing, and other commodities from non-base industries (i.e., induced impacts).

The regional or state-level concentration of ecological restoration firms in places like Oregon [65], North Carolina (see high response count from our survey), Texas, Louisiana, Florida (e.g., [66]), or California (prompting the creation of CalERBA [35]), suggests the potential for mitigation to become a base industry in these regions. Firms now regularly "export" their services across the country, creating banks in multiple states or regions, with income flowing back into their headquarters location. Future work should study if, and how, specific regions around the country have created "hot spots" for mitigation, and ultimately, what could prompt the growth of this strongly "green" industry.

In rare cases, mitigation has been explicitly viewed as an economic development tool, with public investments endeavoring to establish workforces (see Oregon Ecosystem Workforce Program; [67]. Future public investments could expand on these efforts by investing in job training [68] and/or re-training from adjacent industries [69] for the mitigation (or broader restoration) industry and developing supply chains that encourage funds expended on mitigation activities to remain local (i.e., thereby not contributing to the economic base of another region).

Finally, it is important to understand how this study falls into the broader spectrum of industry studies. In most established industries, the information gleaned as part of our surveys would be much more easily available as part of Bureau of Labor Statistics' standard data collection processes from firms previously identified as part of given NAICS-defined industries. Moreover, it would have been collected in a structured fashion that allows more rapid and localized analysis, at county-, regional-, state-, and legislative-district geographic scales. Longer term efforts to track the mitigation industry (as well as the broader restoration sector) would be strongly aided by a formalized (and hierarchical) set of NAICS codes that allow tracking of the firms engaged in purposive ecological restoration, generally, and different facets of mitigation, specifically.

Although widely-accepted "typologies" of ecological restoration do not currently exist (at least not in the United States), we could imagine their use in establishing NAICS codes–many of which may function as secondary codes for many firms–alternatively spanning ecosystem types (wetlands, forests, grasslands), legal drivers (e.g., wetland and stream mitigation, ESA habitat mitigation; [29]), or functional role in establishing restoration sites (e.g., project management [full service mitigation banking, restoration site implementation], legal consulting,

landscaping supplies). Convening and leveraging the input of diverse expert groups–through organizations like the National Research Council (NRC) or the Society for Ecological Restoration (SER)–will be an important step for establishing a comprehensive range of NAICS codes for sustainably describing and tracking the burgeoning mitigation industry and its broader restoration sector.

## Supporting information

**S1 File. Survey modifications and implementation.**
(DOCX)

**S2 File. Survey weighting.**
(DOCX)

**S3 File. Selecting 2014 mitigation firm subset and estimating margins of error.**
(DOCX)

**S4 File. 2021 survey instrument.**
(DOCX)

## Acknowledgments

We would like to thank Becca Madsen (EPIC), Teresa Edwards (UNC Odum Institute, now RTI), and Austin Amandolia (UNC) for their assistance and input. We would also like to thank the 2021 and 2022 ERBA Executive Boards and the ERBA executive director, Sara Johnson.

## Author Contributions

**Conceptualization:** Todd K. BenDor, T. William Lester.

**Data curation:** Todd K. BenDor, Joungwon Kwon.

**Formal analysis:** Todd K. BenDor, Joungwon Kwon, T. William Lester.

**Funding acquisition:** Todd K. BenDor.

**Investigation:** Todd K. BenDor, T. William Lester.

**Methodology:** Todd K. BenDor, T. William Lester.

**Project administration:** Todd K. BenDor.

**Software:** T. William Lester.

**Supervision:** Todd K. BenDor, T. William Lester.

**Validation:** Todd K. BenDor.

**Writing – original draft:** Todd K. BenDor.

**Writing – review & editing:** Todd K. BenDor.

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
