## [Decision Letter · Decision Letter 0]

16 Apr 2023

Assessing the size and growth of the US wetland and stream compensatory mitigation industry

PONE-D-23-05124

Dear Dr. BenDor,

We’re pleased to inform you that your manuscript has been judged scientifically suitable for publication and will be formally accepted for publication once it meets all outstanding technical requirements.

Kind regards,

Muazzam Sabir, Ph.D.

Academic Editor

PLOS ONE

Journal Requirements:

1. Please include your full ethics statement in the ‘Methods’ section of your manuscript file. In your statement, please include the full name of the IRB or ethics committee who approved or waived your study, as well as whether or not you obtained informed written or verbal consent. If consent was waived for your study, please include this information in your statement as well. 

Additional Editor Comments (optional):

Reviewers' comments:

Reviewer's Responses to Questions

**Comments to the Author**

1. Is the manuscript technically sound, and do the data support the conclusions?

Reviewer #1: Yes

Reviewer #2: Yes

2. Has the statistical analysis been performed appropriately and rigorously? 

Reviewer #1: Yes

Reviewer #2: Yes

3. Have the authors made all data underlying the findings in their manuscript fully available?

Reviewer #1: Yes

Reviewer #2: Yes

4. Is the manuscript presented in an intelligible fashion and written in standard English?

Reviewer #1: Yes

Reviewer #2: Yes

5. Review Comments to the Author

Reviewer #1: The study discusses the US Wetland and stream compensatory mitigation market, focusing on the type of firms making up the mitigation "industry", economic impacts and change in industry overtime. Overall, the article is well written and in acceptable form.

Reviewer #2: Authors made good attempt on US wetland and stream compensatory mitigation market, specifically talking about the firms make up the mitigation industry, its economic impacts and change over time. They presented the national survey of mitigation firms and constructed an input-output model of the industry's economic impacts and employment.

6. PLOS authors have the option to publish the peer review history of their article (what does this mean?). If published, this will include your full peer review and any attached files.

Reviewer #1: No

Reviewer #2: No

---

## [Editor Report · Acceptance letter]

22 May 2023

PONE-D-23-05124 

Assessing the size and growth of the US wetland and stream compensatory mitigation industry 

Dear Dr. BenDor:

I'm pleased to inform you that your manuscript has been deemed suitable for publication in PLOS ONE. Congratulations! Your manuscript is now with our production department. 

Kind regards, 

on behalf of

Dr. Muazzam Sabir 

Academic Editor

PLOS ONE